# Strengthening research networks: Insights from a clinical research network in Brazil

**Juliana Freitas Lopes**[1], **Arnaldo Cézar Couto**[2,3], **André Daher**[2], **Bruna de Paula Fonseca**[1] *

1 Center for Technological Development in Health (CDTS), Oswaldo Cruz Foundation (Fiocruz), Rio de Janeiro, RJ, Brazil, 2 Vice Presidency of Research and Biological Collections (VPPCB), Oswaldo Cruz Foundation (Fiocruz), Rio de Janeiro, RJ, Brazil, 3 Rio de Janeiro State University (UERJ), Rio de Janeiro, RJ, Brazil

* bruna.fonseca@fiocruz.br

**Data Availability Statement:** Data cannot be publicly shared because it contains potentially identifying information of human subjects. Data are available from the Fiocruz Ethics Committee

## Abstract

Clinical Research Networks (CRNs) are means to improve healthcare delivery, quality of care and patient outcomes. The Oswaldo Cruz Foundation (Fiocruz), Latin America's leading health research organization, has established a CRN to promote interaction and collaboration among its clinical research experts. After a decade of operation, a revitalization process was undertaken out of the need to improve its functionality. This study aimed to describe the evaluation process of the Fiocruz Clinical Research Network (RFPC) by gathering the opinions and perspectives of its members and identifying the network structure. The goal was to improve scientific collaboration and member engagement, thereby increasing the integration, effectiveness, and impact of clinical research conducted at the institution. Clinical research professionals at Fiocruz were invited to participate in an online questionnaire to collect information about their professional experience, the benefits and constraints of participating in research networks, relevant discussion topics, and the challenges of complying with Good Clinical Practices (GCP). With the help of social network analysis, a deeper understanding of the dynamics and structure of professional communication networks was obtained. The questionnaire was completed by 122 professionals (response rate 50.4%), with most respondents being principal investigators (PIs) with more than 10 years of professional experience (24.6%). Participation in research networks was considered beneficial, particularly in working groups (48.4%), and as an opportunity to exchange experiences with other professionals (44.3%). Almost half of the participants (48.4%) did not identify any barriers to participating in a network. Topics that required further discussion included data management, biorepositories and biobanks, and ethical and regulatory issues. Challenges to conducting clinical research with GCP standards included strategic support and funding, staffing and training, data management, infrastructure, quality management, and collaboration. Communication within the research network was loosely structured, with the most experienced professionals holding central positions. This analysis provided valuable insights to support the management of the RFPC. It highlighted the internal community's interests and expectations, identified key areas for improvement in GCP implementation, and influential professionals who could improve information sharing and national integration. The findings have far-reaching implications that can be applied in different contexts. They

(contact via cepfiocruz@ioc.fiocruz.br) for researchers who meet the criteria for access to confidential data.

**Funding:** Funding was provided by the Oswaldo Cruz Foundation (Fiocruz). Funders had no role in the design of the study, data collection, analysis, interpretation of data or in writing the manuscript.

**Competing interests:** The authors have declared that no competing interests exist.

contribute to the ongoing discussion on the establishment and management of research networks.

## Introduction

Health research networks are often established to improve knowledge flow, facilitate translational science, and shorten pathways from discovery, development and production, to implementation of health technologies [1]. Clinical Research Networks (CRNs) bring together researchers, clinicians, and other healthcare professionals to facilitate collaboration, education and training, clinical trials, implementation research, data sharing, and other research processes [2,3]. The goal of CRNs is to improve healthcare delivery, quality of care, and patient outcomes. CRNs have also been touted as a solution for translating knowledge across research, policy, and practice boundaries [1].

There is a growing need to evaluate and strategically manage research networks to minimize costs, reduce outcome uncertainty, support decision-making, measure key network themes, improve performance, and build social capital [4,5]. Social network analysis (SNA) has been used to provide valuable insights into the structure and dynamics of research networks and to identify key players and opportunities to improve information sharing and collaboration [6]. SNA was used to gain insights into the evolution of scientific connectivity in dengue research in Brazil [7], support intra-organizational network management in tuberculosis research by examining external and internal collaborations [8], and identify key players and map collaborations in a translational research network designed to provide better care for cancer patients [6].

The Oswaldo Cruz Foundation (Fiocruz), part of the Brazilian Ministry of Health, is Latin America's leading health research organization. With more than 11,500 employees nationwide, Fiocruz is involved in all phases of the health industry complex in support of the Brazilian Unified Health System (SUS) to improve the health and quality of life of the Brazilian population and reduce social inequalities [9]. Since 1918, the Foundation has contributed to the development of high-quality clinical research, especially in the field of neglected infectious diseases [10]. Its Clinical Research Platform is responsible for funding and conducting clinical research, besides strengthening the clinical research ecosystem by promoting initiatives such as the Research Ethics Committees Forum, the Biobank Network, the Clinical Trials Monitoring Course, and the Fiocruz Clinical Research Network (RFPC, in Portuguese) [11].

The RFPC was created to promote interaction and collaboration among clinical research professionals at Fiocruz, and to strengthen the strategic role of the institution in this field. Its purpose is to foster a collaborative environment for scientific knowledge production and information exchange among professionals, promoting innovation and excellence in clinical practices, supporting evidence-based decision-making, and thereby expanding the community's technical-scientific capacity. Unlike other research networks, the RFPC is a network of clinical research professionals rather than institutions or individuals interested in disease-specific collaborations. This presents a unique challenge in balancing the different needs from various clinical research areas and provides an important advantage in bridging different perspectives. This ensures that all areas of clinical research benefit from best practices and scientific advances and highlights the importance of professional collaboration across disease-specific boundaries. The governance structure is non-hierarchical and based on the interaction of different bodies such as: Executive Coordination, Consulting Committees, Working Groups, and

Active Members. This approach enables a dynamic and collaborative decision-making process ensuring effective communication and coordination among all stakeholders.

The RFPC was formally established in 2012, with the creation of Communities of Practice covering the intersection of clinical research areas. It has achieved important milestones, including a major annual meeting and regular Communities of Practice meetings, as well as the development of internal communication channels. In 2014, the RFPC formed a Management Committee and prioritized the Fiocruz Biobank Network. In 2015, the Network held two annual meetings and offered several courses. However, starting in 2016, and especially during the COVID-19 pandemic, the productivity of the network began to decline. The network structure became looser and the number of activities decreased. The biggest challenges now are to strengthen scientific collaboration, mobilize the communities of practice, engage existing members, and recruit new members. For this purpose, a comprehensive revitalization plan has been developed with the following phases: i) defining the strategic pillars of the RFPC; ii) benchmarking best practices of network management in the Brazilian context; iii) profiling Fiocruz's clinical research community and identifying their expectations, opportunities, and challenges for participation in the RFPC; and iv) analyzing the internal professional communication network.

In this paper, we focus on the final steps of the RFPC revitalization plan and propose an approach to evaluate the network, according to its structural organization, to inform management and strategic planning. The objective was to outline the RFPC's evaluation process by collecting the opinions and perspectives of its members and identifying the network's structure. The following research questions oriented the analysis: i) What expectations, opportunities, and challenges does the clinical research community anticipate when participating in the RFPC?; ii) How can the internal communication network among Fiocruz clinical research professionals be improved to support the strategic goals of the RFPC?; iii) How can the results of the RFPC assessment be transferred to other research networks to improve their development and management?

## Materials and methods

### Study design

A cross-sectional study was conducted based on i) an online questionnaire to be filled by the Fiocruz clinical research community, containing close and open-ended questions on professional information, benefits and constraints of participating in research networks, and organizational context; ii) an analysis of the social network structure, focusing on the dynamics of professional communication among clinical research professionals.

For this study "clinical research" was defined as studies that necessarily involve human beings, as participants, individually or collectively, directly or indirectly, in whole or in part, and includes the management of information and/or biological material [12]. The study protocol was reviewed and approved by the Fiocruz Research Ethics Committee, number 52560021.2.0000.5248.

### Participants

All Fiocruz clinical research professionals in various positions (principal investigator, research coordinator, monitors, data manager, quality assurance specialist etc.), were invited to respond to the online questionnaire. All units of Fiocruz throughout Brazil (Fig 1) were included. Fiocruz headquarters in Rio de Janeiro include 12 different technical-scientific units: Evandro Chagas National Institute of Infectious Diseases (INI); Fernandes Figueira National Institute for Women's, Children's and Adolescent Health (IFF); Vice-Presidency (VP); Oswaldo Cruz

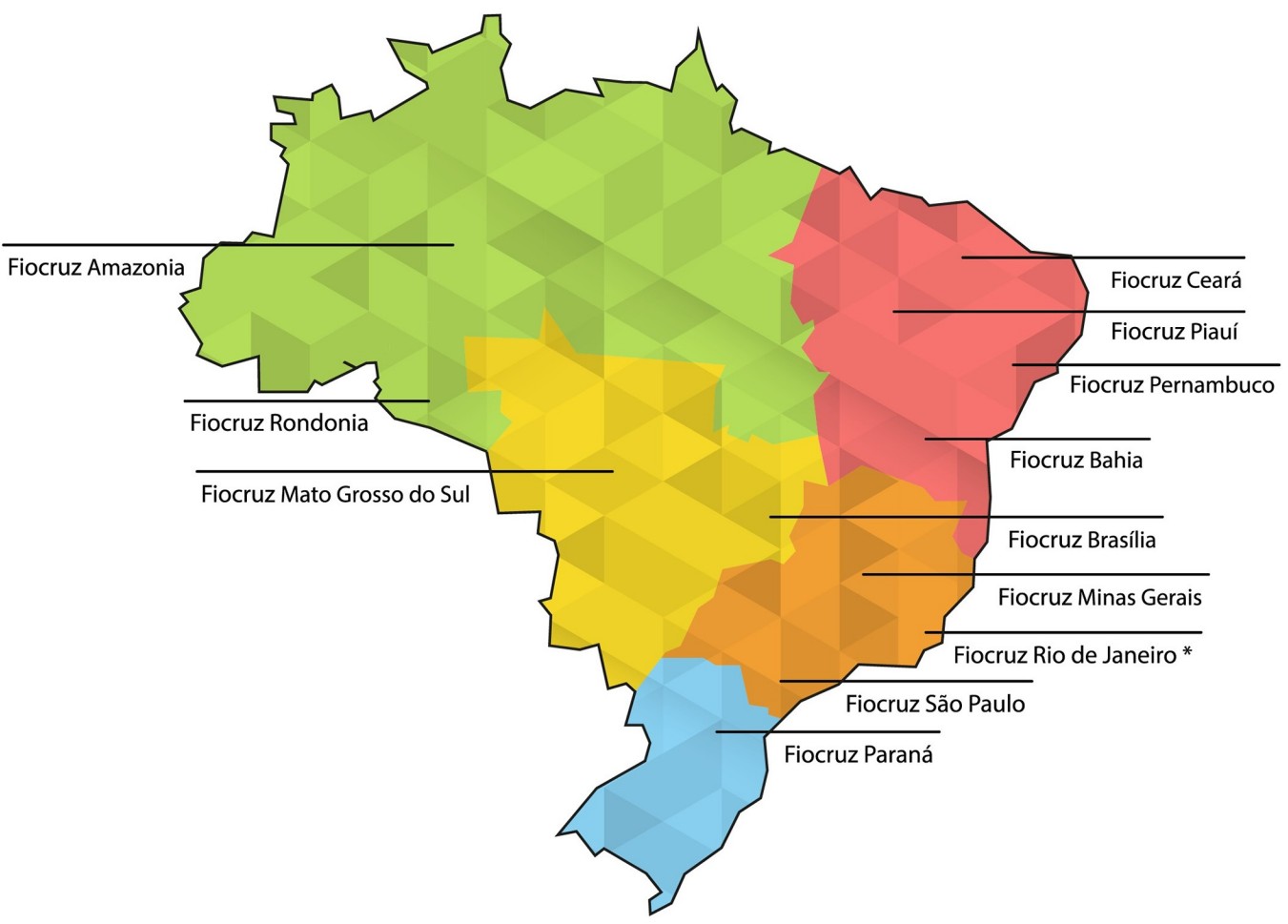

**Fig 1. Fiocruz technical-scientific units participating in the study.** *Fiocruz Rio de Janeiro has 12 different technical-scientific units. Colors indicate geographical regions of Brazil: North (green); Northeast (red); Midwest (yellow); Southeast (orange); South (blue). Map of Brazil obtained from https://mapsvg. com/maps/brazil, used under a CC BY 4.0 license, and modified from the original to include indications of Fiocruz units and geographical regions.

Institute (IOC); National School of Public Health (ENSP); Scientific Computing Program (PROCC); National Institute for Quality Control in Health (INCQS); Covid-19 Diagnostic Support Unit (UNADIG); Institute of Immunobiological Technology (Bio-Manguinhos); Institute of Drug Technology (Farmanguinhos); Center for Technological Development in Health (CDTS); Institute of Scientific and Technological Communication and Information in Health (ICICT).

## Construction of the questionnaire

The questionnaire consisted of 16 questions divided into four sections:

i) Professional information: role in clinical research, place of work (technical-scientific unit), and years of experience in clinical research at Fiocruz. A question about conducting research involving human subjects was included to confirm that the respondent was directly or indirectly involved in clinical projects.

ii) Benefits and constraints: two close-ended questions on expected benefits of participating in research networks (choose up to 3 answers out of 10 options) and potential barriers (choose up to 3 answers out of 7 options).

iii) Organizational context: two open-ended questions to identify relevant discussion topics in clinical research and challenges in conducting clinical research under Good Clinical Practices (GCP) standards at Fiocruz.

iv) Internal communication network: using a "name generator" question [13] respondents were asked to name at least three other clinical research professionals working at Fiocruz with whom they would discuss project-related ideas or seek support.

Responses to the question on the challenges of conducting clinical research under GCP standards were reviewed by two independent researchers and grouped into the following categories: strategic support and funding; human resources and training; data management; infrastructure; quality management; and collaboration.

The Research Electronic Data Capture (REDcap) platform [14] was used to create the questionnaire and host the response database under the supervision of a data scientist to ensure the accuracy, security, and reliability of the information retrieved.

The open invitation was sent to all Fiocruz clinical research professionals through an institutional mailing list via secure web link (available from November 11, 2021, to March 25, 2022). Emails of Fiocruz researchers formally involved in clinical trials were also retrieved from the Brazilian Clinical Trials Registry (Rebec) and the ClinicalTrials.gov platform. Reminder emails were sent in the following months to maximize the response rate. Before participants were allowed to access the questions, they were given an online Informed Consent Form (ICF) indicating their agreement to participate in the study.

## Dimensions and forms of analysis

Four dimensions oriented the analysis: professional characterization, benefits and constraints, organizational context, and internal communication networks (Fig 2).

MS Excel was used to calculate simple frequencies and percentages of close-ended questions. The whole text of open-ended responses was imported into the VantagePoint software (Search Technologies Inc.) to create a word cloud of the most frequently cited words. Articles, prepositions, and common words such as "research" and "clinic" were excluded. Related terms such as "data analysis", "data privacy" and "data handling" were grouped under "data management". In the image, the size of a word is proportional to how often they appeared in the text.

The Gephi software v.9.3 [15] was used to create, visualize, and analyze the internal communication network of clinical research professionals. In the network, each professional was represented by a circle (node), and the links between them were represented by directional arrows. Arrows were directed from the respondent to the person he/she reported as primary contacts.

Gephi and UCINET [16] software were used for statistical analysis. The following metrics were used to describe the structural patterns of the communication flow [17]:

i) Number of nodes: number of professionals within the network.

ii) Number of links: number of connections between professionals within the network.

iii) Size of the giant component: the largest group of professionals connected through their work relationships. The larger the giant component, or the proportion of professionals it contains, the more interconnected the network is.

iv) Diameter: the maximum distance between all possible pairs of nodes. It is the length of the longest path connecting two professionals in the network. In a communication network, the diameter indicates how quickly information reaches everyone in the network.

v) Density: is the ratio between the number of existing connections and the total number of possible connections between professionals. Network density ranges from 0 to 100, with higher values indicating more frequent communication between all members of the network.

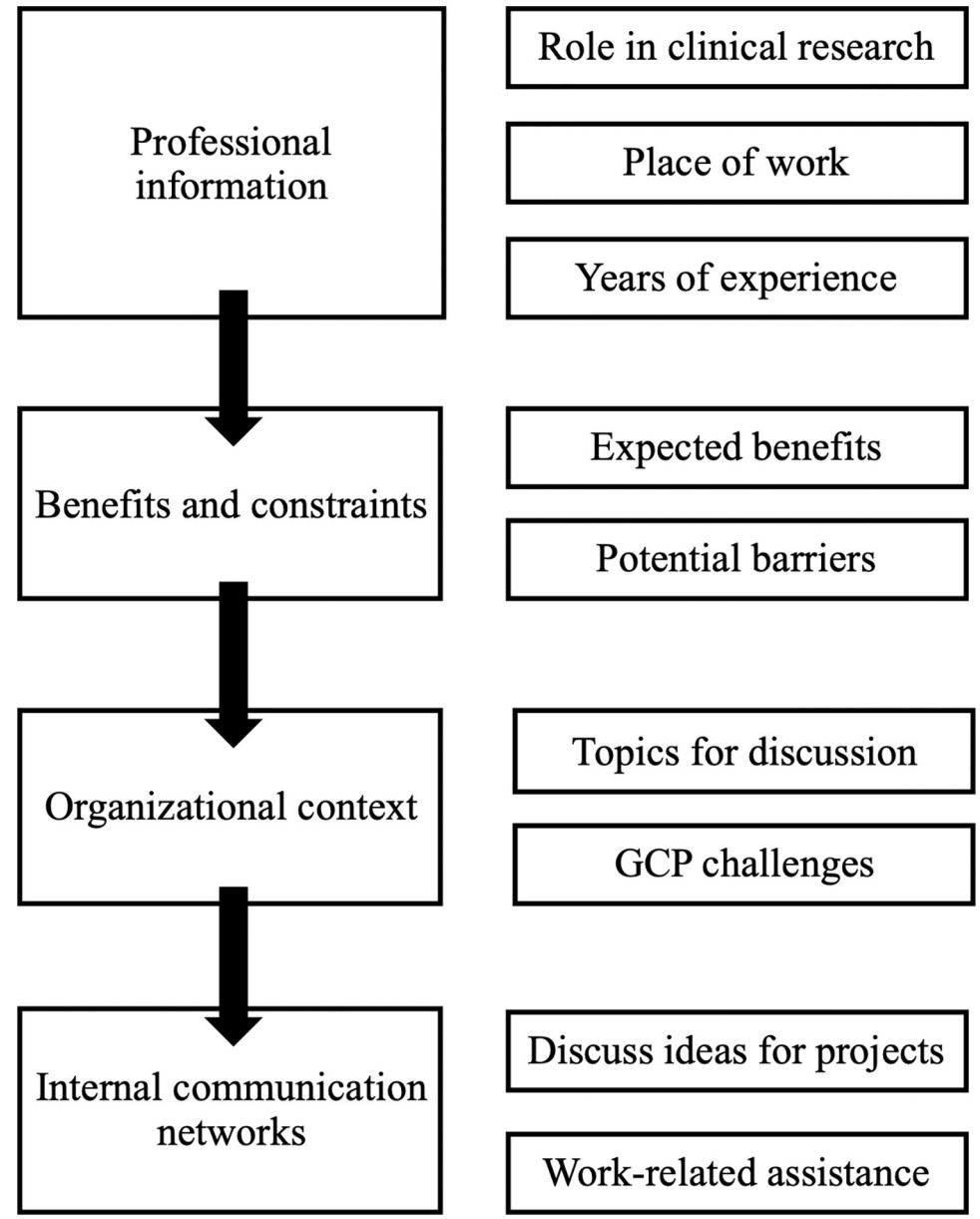

**Fig 2. Analysis model.**

v) External-internal index (E-I index): calculates the level of internal relationships (homophilia) by measuring the number of connections with members of the same group (internal) compared to connections with members of another (external) group. The index value varies from -1 (all connections are internal to the group) to 1 (all connections are external to the group). The lower the E-I index of a network, the more endogenous the relationships. In this study, each group represented a technical-scientific unit of Fiocruz, some of them located in different geographical locations (Fig 1).

Influential professionals in the network were identified based on their indegree centrality, which reflects the total number of links directed to them [18]. An individual is influential because he or she receives communication from many other network members due to his/her

function in the organizational hierarchy or reputation as a valuable source of information. In this study, indegree centrality expresses the number of people who indicated a particular professional as a contact for support and discussion of ideas. High indegree centrality indicates more direct connections to many members of the network, making that person a focal point of communication.

## Results

The questionnaire was accessed by 242 individuals, approximately 80% of the Fiocruz clinical research community. Of these, 49 individuals were excluded from the analysis as they did not accept the ICF, and 51 individuals accepted the ICF but did not provide any answers to the questionnaire. Of the remaining 142 participants, 20 duplicates were removed, and the last response was considered valid. A total of 122 (50.4%) valid responses were included in the final analysis (Fig 3).

### Professional information

The respondents represented various technical-scientific units of Fiocruz, with a notable concentration from Rio de Janeiro—INI (37.7%), followed by IOC (13.1%), Bio-Manguinhos (9.8%), and IFF (9.8%).

Table 1 presents the distribution of respondents based on their role in clinical research and years of experience. Most participants (46.2%) reported having more than 10 years of professional experience, the Principal Investigators (PIs) accounting for 24.6% of those (Table 1).

### Expected benefits and constraints of participating in research networks

The most frequently mentioned benefit was the opportunity to participate in working groups on topics relevant to Fiocruz's clinical research (48.4%), and 44.3% mentioned interest in sharing experiences and seeking support from other professionals (Table 2). Most participants (48.4%) did not identify any specific barriers to joining a research network. However, 36.9% cited lack of time for activities outside of their daily commitments as a limitation (Table 2).

### Organizational context

The word cloud (Fig 4) illustrates what respondents considered important clinical research topics for discussion. The most frequently mentioned topics were data management, biorepositories and biobanks, ethics and regulatory issues, funding, and clinical trial management and monitoring.

Six key issues emerged as the biggest obstacles for Fiocruz to conduct clinical research by Good Clinical Practice (GCP) standards: Strategic support and funding; Human resources and training, Data management; Infrastructure; Quality management, and Collaboration (Table 3).

### Internal communication network

The social relationship structure of the RFPC was analyzed by mapping work-related interactions (Fig 5). The communication network represents the professional relationships that exist outside of formal agreements between technical-scientific units and can be described as informal relationships based on trust.

The network was created using 59% of the valid responses (72 out of 122), where circles represent individual clinical research professionals and links between them represent direct communication (Fig 5). These links were directed from the respondent to the individual identified

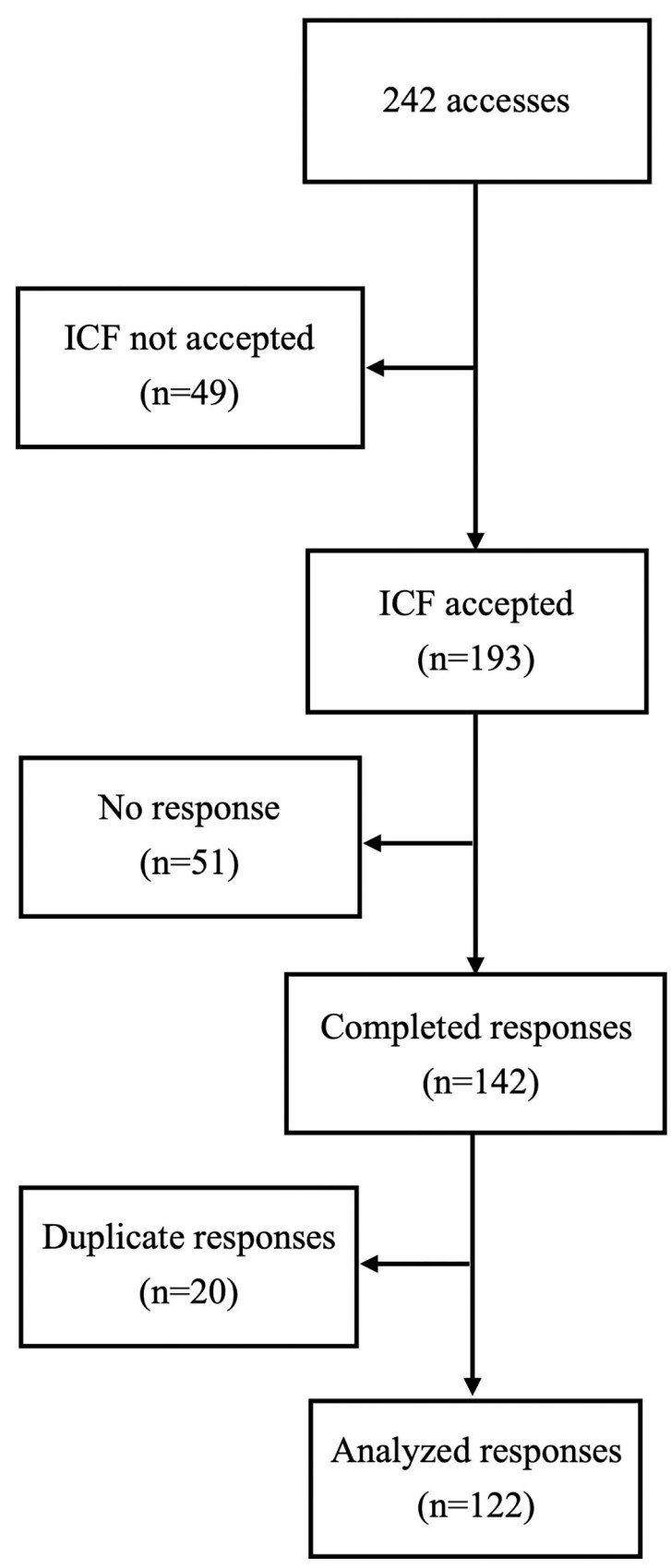

**Fig 3. Sampling flowchart.**

**Table 1. Respondents' role in clinical research and years of experience at Fiocruz.**

| Role in clinical research | N (%) | Years of experience | N (%) |
|---|---|---|---|
| Principal investigator | 46 (37.7%) | < 1 | 2 (1.6%) |
| | | 1–2 | 1 (0.8%) |
| | | 2–5 | 2 (1.6%) |
| | | 5–10 | 11 (9.0%) |
| | | > 10 | 30 (24.6%) |
| Clinical research coordinator | 21 (17.2%) | < 1 | 1 (0.8%) |
| | | 1–2 | 1 (0.8%) |
| | | 2–5 | 1 (0.8%) |
| | | 5–10 | 9 (7.4%) |
| | | > 10 | 9 (7.4%) |
| Study physician | 14 (11.5%) | 1–2 | 1 (0.8%) |
| | | 2–5 | 3 (2.5%) |
| | | 5–10 | 1 (0.8%) |
| | | > 10 | 9 (7.4%) |
| Clinical research monitor | 8 (6.6%) | < 1 | 4 (3.3%) |
| | | 2–5 | 2 (1.6%) |
| | | 5–10 | 2 (1.6%) |
| Data manager | 4 (3.3%) | 5–10 | 2 (1.6%) |
| | | > 10 | 2 (1.6%) |
| Quality assurance specialist | 4 (3.3%) | < 1 | 1 (0.8%) |
| | | 5–10 | 2 (1.6%) |
| | | > 10 | 1 (0.8%) |
| Clinical research assistant | 3 (2.5%) | < 1 | 1 (0.8%) |
| | | 2–5 | 1 (0.8%) |
| | | 5–10 | 1 (0.8%) |
| Regulatory affairs | 3 (2.5%) | < 1 | 1 (0.8%) |
| | | 5–10 | 1 (0.8%) |
| | | > 10 | 1 (0.8%) |
| Collaborator researcher | 5 (4.1%) | < 1 | 1 (0.8%) |
| | | 2 to 5 | 1 (0.8%) |
| | | 5–10 | 1 (0.8%) |
| | | > 10 | 2 (1.6%) |
| Research nurse | 2 (1.6%) | 1–2 | 1 (0.8%) |
| | | 5–10 | 1 (0.8%) |
| Pharmacist | 2 (1.6%) | < 1 | 1 (0.8%) |
| | | 5–10 | 1 (0.8%) |
| Project manager | 2 (1.6%) | 5–10 | 1 (0.8%) |
| | | > 10 | 1 (0.8%) |
| Laboratory technician | 1 (0.8%) | 5–10 | 1 (0.8%) |
| Data analyst | 1 (0.8%) | 5–10 | 1 (0.8%) |
| Drug safety specialist | 1 (0.8%) | > 10 | 1 (0.8%) |
| Other | 5 (4.1%) | < 1 | 1 (0.8%) |
| | | 1–2 | 1 (0.8%) |
| | | 5–10 | 2 (1.6%) |
| | | > 10 | 1 (0.8%) |

**Table 2. Benefits and constraints of participating in a research network.**

| Benefits | | Constraints | |
|---|---|---|---|
| Participation in working groups on topics relevant to clinical research at Fiocruz | 48.4% | None | 48.4% |
| Share experiences and seek assistance from other professionals | 44.3% | Lack of time for activities other than usual | 36.9% |
| Obtain funding for research projects | 39.3% | Lack of support from my unit | 10.7% |
| Training activities | 32.0% | Negative experiences with other clinical research networks | 4.1% |
| Technical/scientific support for conducting my research according to GCP | 28.7% | Participation in similar networks | 4.1% |
| Networking | 27.9% | Lack of interest in participating in research networks/ poor attractiveness of the network | 2.5% |
| Up-to-date information on clinical research | 25.4% | Don't know | 0.8% |
| Improve my ability to design more competitive projects | 11.5% | | |
| Get assistance in developing research protocols | 5.7% | | |

as primary contact. The size of the nodes corresponds to their indegree centrality, which indicates how often they were identified as contacts.

The RFPC communication network included 151 clinical research professionals, with a giant component of 137 members, representing approximately 90.7% of the network (S1 Table). The large diameter and low density of the network indicate a loose structure, suggesting a possible lack of familiarity between professionals or limited coordination among them (S1 Table).

INI, the main unit for clinical research and healthcare, has the highest representation in the network with 36.4% of professionals (n = 55) (Fig 5). Next is the IOC with 15.2% of professionals (n = 23), which focuses on basic research and technology development, providing diagnostic services for infectious and genetic diseases. The office of the Vice-President (VP) and Bio-Manguinhos each account for 7.9% of professionals (Fig 5). The presence of isolated groups at the periphery of the network suggests a possible lack of interaction or limited integration into the wider community.

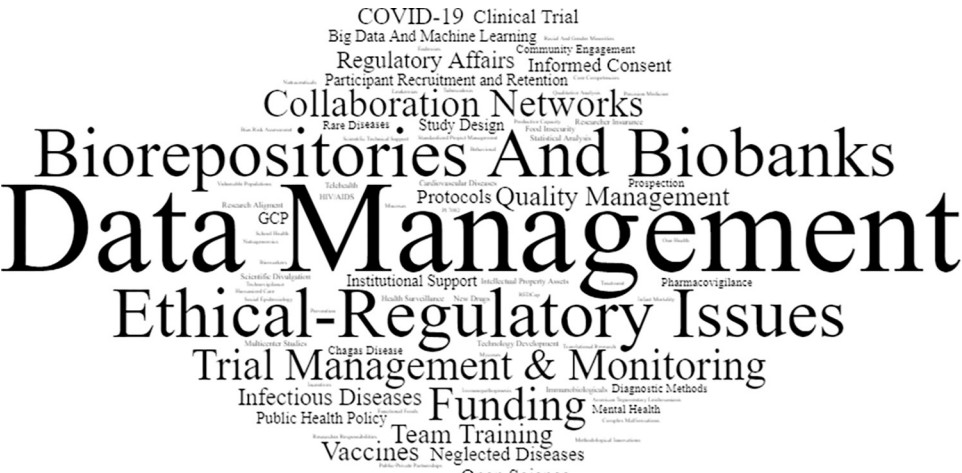

**Fig 4. Word cloud of clinical research topics for discussion at Fiocruz.**

**Table 3. Challenges in conducting research under GCP standards within Fiocruz.**

| Topics | Verbatim responses |
|---|---|
| Strategic support and funding | *"Build a strategic plan for clinical research with expanded support for researchers."* *"Fund research and hire qualified professionals."* |
| Human resources and training | *"High turnover of professionals requiring continuous training."* *"Motivate researchers to educate themselves as well as their team."* |
| Data management | *"Have an open database for clinical research protocols."* *"Lack of knowledge about precision medicine and its individual and collective clinical applications data management."* |
| Infrastructure | *"Increase investment in infrastructure to meet international quality standards in clinical research."* *"Lack of infrastructure for documenting biological samples and storage."* |
| Quality management | *"Lack of quality assurance systems."* *"Institutionalize quality processes."* |
| Collaboration | *"Poor integration of non-clinical & clinical researcher."* *"Poor professional communication and collaboration between Fiocruz units."* |

The five most central professionals in the RFPC communication network were clinical research coordinators with 5 to 10 years of experience (P01 and P04) and principal investigators with more than 10 years of clinical research experience at Fiocruz (P02, P03, and P05) (Fig 5). Four of them were affiliated to INI, and one to ENSP (P03). Their high indegree centrality indicated that their colleagues would seek them out to solve work-related challenges or discuss ideas, which showed they were approachable and had valuable expertise.

Analysis of the communication network of the 20 technical-scientific units participating in the study showed that half of them considered the INI as the main point of contact for work-related support and project discussions (S1 Fig). IOC and VP followed closely with seven and six units respectively (S1 Fig).

In general, Fiocruz units had more internal relationships (n = 113) than external relationships (n = 74), resulting in an E-I index of -0.209. In particular, INI mainly maintains internal relationships (E-I = 0.584), although other units often seek its support. In contrast, VP and IOC units tended to have more external relationships (S2 Table).

## Discussion

The RFPC plays an important role in strengthening the interaction between the clinical research professionals at Fiocruz. The results presented here provided an overview of the profile, needs and interests of this community. Most respondents had more than 10 years of professional experience and expressed strong interest in participating in the RFPC, especially as part of working groups on important clinical research topics. Time constraints were cited as a barrier to active participation in the network. Data management was a key area of interest and was also cited as a challenge for GCP compliance. Analysis of the communication network revealed a loose structure with clinical research coordinators and principal investigators as central figures. In most Fiocruz units internal relationships predominated, especially within the INI.

Network management involves overseeing all aspects of network formation and operation. This includes developing concepts, identifying appropriate members and collaborators, organizing and establishing networks, creating services and programs, and developing communication and monitoring strategies [19]. Building collaborative networks in the health sector has been shown to transform inefficiencies and poor outcomes in a variety of scenarios [20,21].

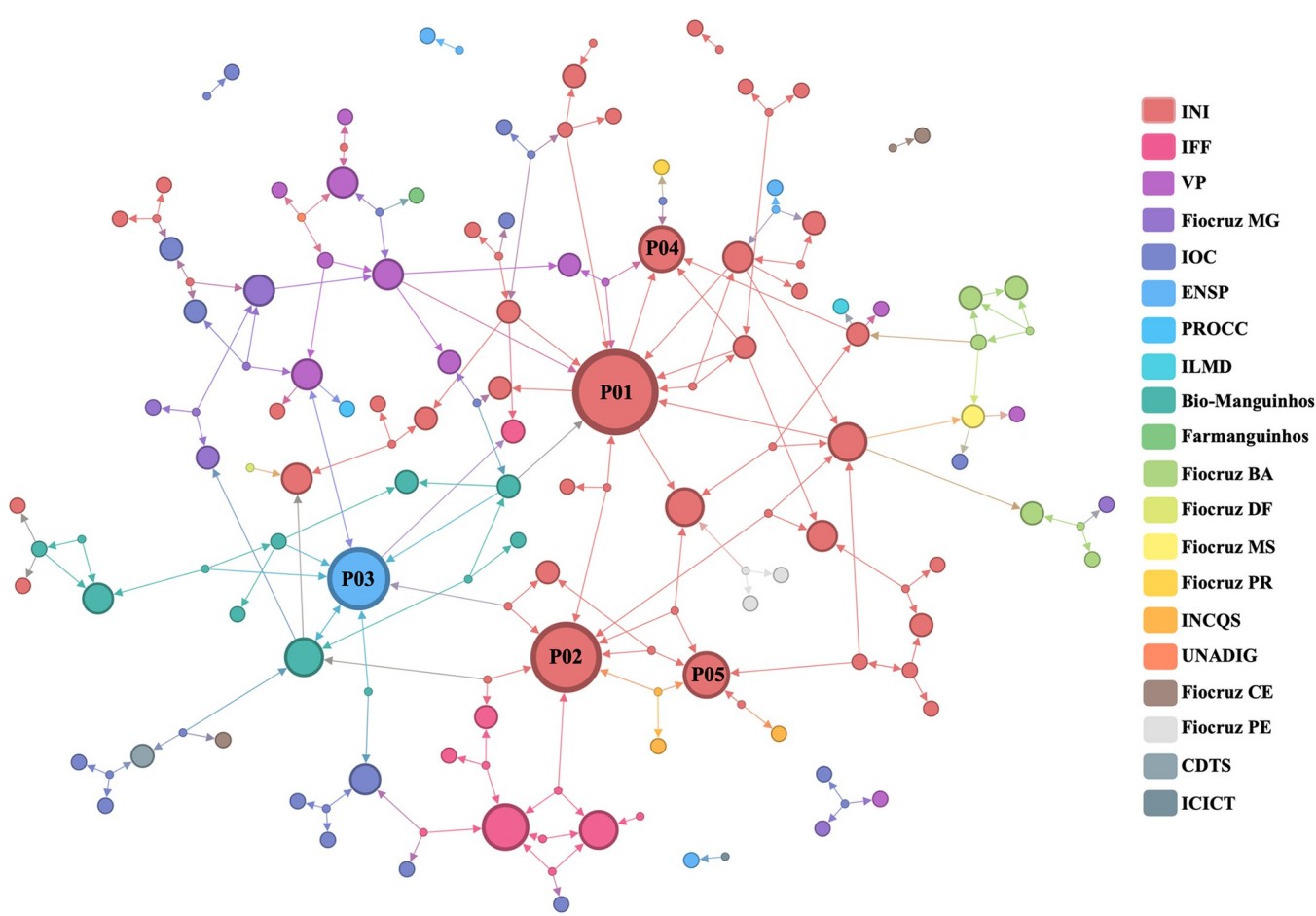

**Fig 5. Fiocruz internal communication network in clinical research.** Circles (nodes) represent professionals, and their links represent direct communications between respondents and their primary contacts. The size of a node is proportional to its indegree centrality and node colors represent their technical-scientific units. P01 to P05 are the most central professionals in the network. Evandro Chagas National Institute of Infectious Diseases (INI); Fernandes Figueira National Institute for Women's, Children's and Adolescent Health (IFF); Vice-Presidency (VP); Oswaldo Cruz Institute (IOC); National School of Public Health (ENSP); Scientific Computing Program (PROCC); National Institute for Quality Control in Health (INCQS); Covid-19 Diagnostic Support Unit (UNADIG); Institute of Immunobiological Technology (Bio-Manguinhos); Institute of Drug Technology (Farmanguinhos); Center for Technological Development in Health (CDTS); Institute of Scientific and Technological Communication and Information in Health (ICICT); Fiocruz Minas Gerais (Fiocruz MG); Fiocruz Mato Grosso do Sul (Fiocruz MS); Fiocruz Paraná (Fiocruz PR); Fiocruz Bahia (Fiocruz BA); Fiocruz Ceará (Fiocruz CE); Fiocruz Pernambuco (Fiocruz PE); Fiocruz Distrito Federal (Fiocruz DF); Fiocruz Amazonas (Fiocruz AM).

Many organizations have established internal networks to facilitate communication and learning across different work functions, units, levels, and departments [22,23]. These networks play a key role in helping organizations identify people and resources to effectively address problems, while also providing a platform for employees to communicate and learn from each other [24]. Ten years after the RFPC was established, a revitalization process became essential to review progress and improve function and efficiency.

The results presented here are a testament to the diverse clinical research expertise at Fiocruz. The high level of participation in the survey by experienced professionals demonstrates their great interest in the RFPC, which in turn enhances the reliability of the network. These professionals bring valuable knowledge, expertise, and skills to ensure adherence to strict scientific standards, regulatory compliance, participant safety, and achievement of reliable and meaningful research results. In addition, their experience enables them to serve as mentors and build collaborations within the research community, thereby facilitating access to

resources, funding opportunities, and interdisciplinary expertise. As part of the goal of revitalizing the RFPC, it is also important to develop strategies to include other professionals who may be underrepresented in the survey.

To ensure the sustainability and effectiveness of networks, it is essential to collect information on the key contributing factors for their development and productivity, and the main challenges of maintaining such networks [25]. The survey found that the benefits of participating in RFPC outweighed the challenges. However, managers must be aware of the time constraints associated with these actions and that they must be supported by their supervisors. Previous studies have identified "human factors" as being a priority for developing and maintaining a collaborative research network, including leadership and a shared vision [26].

To remain current and relevant, networks should stay abreast of emerging research topics arising from new technologies and methodologies. These advances provide opportunities to develop engagement strategies, such as forming working groups, organizing regular workshops, and hosting discussion forums on selected topics. These strategies are part of a broader clinical research capacity-building initiative that is essential for governments and health authorities, particularly in developing countries [27].

Building capacity for GCP in research and clinical trials remains a global challenge [28]. At Fiocruz, survey respondents identified various barriers to conducting clinical research following GCP standards. These challenges reflect the difficulties in implementing GCP in developing countries. Limited resources may hinder investment in training and infrastructure, while a lack of GCP training may result in a poor understanding of research ethics, participant protection, data management, and quality assurance. In addition, limited regulatory frameworks and capacity may lead to inconsistent practices and potential ethical issues [29]. The RFPC must identify these factors in order to develop and implement a strategic plan to address these gaps and continuously improve the quality of clinical research at Fiocruz [27].

SNA provided valuable insights into communication patterns within the research network, enabling management to propose informed solutions. Knowledge exchange occurs primarily through informal social communication networks, which are often very different from formal organizational structures and work processes [30]. Regardless of the size of the organization, close connections between all professionals are rare, especially in national institutions. The results suggest that management should promote opportunities for stronger professional interaction. Several strategies can help RFPC accomplish this task, such as: allocating time and resources for collaboration, creating an organizational environment that encourages the exchange of ideas and perspectives, organizing face-to-face meetings and forums for sharing, bridging disciplinary boundaries through evidence-based principles and strategies, and developing project management and communication skills [31–33].

The greatest influence of experienced professionals highlights the key role of seniority in the Fiocruz clinical research community. This "preferential connection" to experienced experts is well-documented across networks [34] and has also been observed in other Fiocruz networks [8]. These professionals can make a significant contribution to the revitalization and dissemination of new RFPC by actively participating in steering committees, acting as catalysts for knowledge transfer, and providing valuable leadership and guidance [19]. This has been demonstrated in other healthcare networks, where central players have helped strengthen collaboration among members [6,35].

The relationship between technical-scientific units is centered around three key units: INI, IOC and VP, all located in the city of Rio de Janeiro. As multidisciplinary is a catalyst for successful innovation [36], interaction and collaboration between other units must be encouraged to optimize project implementation and avoid duplication of resources. For example, exploring commonalities between diseases through research networks has been shown to enable

multiple groups to address a broad range of common problems [21]. Institutional incentives such as the promotion of specialized courses, workshops, conferences, and the development of joint projects for specific funding calls can be an effective means to promote and facilitate such collaborations.

The E-I index shows that the number of internal relationships in Fiocruz units is greater than external relationships leading to the clustering of individuals and units. It is well known that collaborative knowledge generation is affected by various "distances" [37]. These distances include spatial proximity (geographic distance), similarity of knowledge bases (cognitive distance), and belonging to the same part of the organization (organizational distance). These factors facilitate communication and knowledge sharing, and ultimately reduce barriers to collaboration [37]. By implementing strategies to promote collaboration through RFPC, Fiocruz can improve the diffusion of productive collaborative interactions.

### Strengths and limitations

Strengths of our study include the diverse representation of clinical research professionals, which provided a wide range of perspectives. The use of open and closed-ended questions allowed participants to fully express their opinions, and their perspectives were respected. Our analytical model can be replicated for regular assessment and monitoring of the development of the network, and the approach can also be adapted to study other areas of knowledge. These factors demonstrate the robustness and flexibility of our study, which provides relevant data for the current context and is a valuable tool for future research and expansion of network analysis in various scientific fields.

We recognize that while the number of participants was relatively large, they may not be representative of all clinical research professionals at Fiocruz. We anticipate that future studies facilitated by the establishment of the RFPC based on our results, will have a larger number of participants. Our network analysis relied on participants naming professionals to form the collaborative network. The open-ended nature of the questions may have resulted in an inability to fully recall all professional connections, a common limitation of network analyses without predefined options. As Fiocruz's headquarters in Rio de Janeiro have the highest concentration of clinical research professionals, our results primarily reflect this community. We anticipate that future studies will achieve a more balanced representation of all Fiocruz units as the RFPC continues to grow.

### Conclusions

To strengthen the strategic role of Fiocruz clinical research, improved interactions between professionals are needed. A modern strategic organization that wants to excel and optimize resources, time, and outputs must be rooted in a collaborative culture. This study serves as a preliminary analysis to support the RFPC revitalization plan. The recommendations derived from the network analysis will increase the integration of the network in the future. Ongoing monitoring and evaluation will ensure that the interests of the Fiocruz clinical research community are considered. In addition, follow-up studies will provide a longitudinal perspective on the progress of collaborative efforts.

### Supporting information

**S1 Fig. Interactions between Fiocruz technical-scientific units.** The circles (nodes) represent technical-scientific units, and the links between them represent a direct relationship from the respondent's technical-scientific unit to his/her primary contacts. The size of a node is proportional to its indegree centrality. Auto-loops indicate that some professionals have identified a

colleague from the same unit as their primary contact. Evandro Chagas National Institute of Infectious Diseases (INI); Fernandes Figueira National Institute for Women's, Children's and Adolescent Health (IFF); Vice-Presidency (VP); Oswaldo Cruz Institute (IOC); National School of Public Health (ENSP); Scientific Computing Program (PROCC); National Institute for Quality Control in Health (INCQS); Covid-19 Diagnostic Support Unit (UNADIG); Institute of Immunobiological Technology (Bio-Manguinhos); Institute of Drug Technology (Farmanguinhos); Center for Technological Development in Health (CDTS); Institute of Scientific and Technological Communication and Information in Health (ICICT); Fiocruz Minas Gerais (Fiocruz MG); Fiocruz Mato Grosso do Sul (Fiocruz MS); Fiocruz Paraná (Fiocruz PR); Fiocruz Bahia (Fiocruz BA); Fiocruz Ceará (Fiocruz CE); Fiocruz Pernambuco (Fiocruz PE); Fiocruz Distrito Federal (Fiocruz DF); Fiocruz Amazonas (Fiocruz AM). (TIF)

**S1 Table. Metrics of the RFPC communication network.** (DOCX)

**S2 Table. E-I index for Fiocruz technical-scientific units.** (DOCX)

## Acknowledgments

We thank Dr. Priscila Costa Albuquerque for assistance with the visualization of results and Dr. Fabio Zicker for the critical review of earlier versions of this manuscript.

## Author Contributions

**Conceptualization:** Juliana Freitas Lopes, Arnaldo Cézar Couto, Bruna de Paula Fonseca.

**Data curation:** Juliana Freitas Lopes, Bruna de Paula Fonseca.

**Formal analysis:** Juliana Freitas Lopes, Bruna de Paula Fonseca.

**Funding acquisition:** Arnaldo Cézar Couto, André Daher.

**Methodology:** Bruna de Paula Fonseca.

**Project administration:** Bruna de Paula Fonseca.

**Supervision:** André Daher, Bruna de Paula Fonseca.

**Validation:** André Daher.

**Writing – original draft:** Juliana Freitas Lopes.

**Writing – review & editing:** Arnaldo Cézar Couto, André Daher, Bruna de Paula Fonseca.

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
