## [Decision Letter · Decision Letter 0]

4 Jun 2024

PONE-D-23-39952Strengthening research networks: insights from a nationwide clinical research network in BrazilPLOS ONE

Dear Dr. Fonseca,

Thank you for submitting your manuscript to PLOS ONE. I am happy to send you the Reviewer Comments and apologize again for the time in review.  After careful consideration, we feel that it has merit but does not fully meet PLOS ONE’s publication criteria as it currently stands. Therefore, we invite you to submit a revised version of the manuscript that addresses the points raised during the review process. 

We look forward to receiving your revised manuscript.

Kind regards,

Claudia Garcia Serpa Osorio-de-Castro, Ph.D

Academic Editor

PLOS ONE

2. In the online submission form you indicate that your data is not available for proprietary reasons and have provided a contact point for accessing this data. Please note that your current contact point is a co-author on this manuscript. According to our Data Policy, the contact point must not be an author on the manuscript and must be an institutional contact, ideally not an individual. Please revise your data statement to a non-author institutional point of contact, such as a data access or ethics committee, and send this to us via return email. Please also include contact information for the third party organization, and please include the full citation of where the data can be found.

Reviewers' comments:

Reviewer's Responses to Questions

**Comments to the Author**

1. Is the manuscript technically sound, and do the data support the conclusions?

Reviewer #1: Yes

Reviewer #2: Yes

2. Has the statistical analysis been performed appropriately and rigorously? 

Reviewer #1: Yes

Reviewer #2: No

3. Have the authors made all data underlying the findings in their manuscript fully available?

Reviewer #1: No

Reviewer #2: No

4. Is the manuscript presented in an intelligible fashion and written in standard English?

Reviewer #1: Yes

Reviewer #2: Yes

5. Review Comments to the Author

Reviewer #1: The article presents a relevant topic and makes a contribution to network analysis. The materials and methods section requires revision, as the order is not in a logical chronology. It is suggested to start with the study design, participants, construction of the questionnaire, dimensions analysis and forms of analysis.

Reviewer #2: General comments

This paper assessing a clinical research network in Brazil is interesting, given the importance to describe and evaluate such networks. Parts of the analyses are mainly of local/regional interest, while others are generalizable. I believe, however, that there is room for improvement in the context description and discussion to add value for readers in other settings.

Specific comments for revision

a) Major

Abstract: Number of respondents and response rate is not presented in the abstract

Abstract & introduction: I lack a clearly formulated aim, both in the abstract and in the end of the introduction section. The message is there, but it could be described more clear, also potentially including a few specific research questions.

Introduction: It could be further stressed how this network benefits both research and healthcare. It would also be valuable with more comparisons with similar networks in other settings. What are the key features of this network and how does it differ from others regarding management, staffing, aims, structure, and the way how it integrates academy with clinical care.

Introduction: I is claimed that they started some ten years ago, and such networks take time to develop. It would be valuable to get some description of the time line, and important milestones.

Methods :Page 9-10 incl Fig 2. This description of the population is very valuable but should come earlier, preferably first in the methods section directly as setting or population. It is also possible to place it in the introduction together with a more comprehensive description of the network.

Methods/results: No statistical measure of uncertainty has been applied, e.g. adding confidence intervals on proportions and comparisons. Overall I cant see that many comparisons between groups. Perhaps this is wise given the small numbers, but it might have been interesting to see differences in responses based on respondent categories (with an appropriate statistical test)

Results: The initial sentence “questionnaire was accessed by”, does it mean that this number was invited?

Results: There are many illustrations – in total 6 figures and 5 tables. I don’t know the maximum allowed number by the journal, but suggest some could be moved to appendices

Discussion: The structure could be improved through adding a first sharp paragraph about the key findings, then more integration between the assessment of own findings in relation to others. Overall, the generalizability and and comparisons with what is known in the scientific literature should be emphasized more to increase the value for the global scientific community.

Discussion: There is a lack of a Strengths and weaknesses section, which is a necessary part of a discussion.

b) Minor

There are no subheadings in Abstract (Intro, Aim, Methods, Result, Discussion). I don’t know whether it is required by the journal or not, but think it would add value.

Texts: Some subheaders have different sizes. There is also some unnecessary repetition in the text (e.g. the response rate that comes twice) and quite a lot of numbers presented in double both in a table and in the free text. The results and discussion parts could therefore be revised and shortened slightly. I also recommend some statements to be modified to emphasize they are perceived by responders.

Table 5 –Abbreviations should be explained in footer

6. PLOS authors have the option to publish the peer review history of their article (what does this mean?). If published, this will include your full peer review and any attached files.

Reviewer #1: No

Reviewer #2: No

---

## [Author Response · Author response to Decision Letter 0]

8 Jul 2024

We thank the reviewers for their interest and useful comments on the manuscript.

REVIEWER 1

Comment: The article presents a relevant topic and makes a contribution to network analysis. The materials and methods section requires revision, as the order is not in a logical chronology. It is suggested to start with the study design, participants, construction of the questionnaire, dimensions analysis and forms of analysis.

Response: The methodology is now in a logical chronological order, as suggested. 

REVIEWER 2

Comment: This paper assessing a clinical research network in Brazil is interesting, given the importance to describe and evaluate such networks. Parts of the analyses are mainly of local/regional interest, while others are generalizable. I believe, however, that there is room for improvement in the context description and discussion to add value for readers in other settings.

Response: Context description and discussion were improved to add value to the paper, as suggested. See the following comments.

Comment: Abstract: Number of respondents and response rate is not presented in the abstract

Response: The information was included, as suggested. See lines 31 and 32.

Comment: Abstract & introduction: I lack a clearly formulated aim, both in the abstract and in the end of the introduction section. The message is there, but it could be described more clear, also potentially including a few specific research questions.

Response: The aim of the study is now clearer and included in the abstract and introduction, as suggested. See lines 21 to 26 and 121 to 128

Comment: Introduction: It could be further stressed how this network benefits both research and healthcare. It would also be valuable with more comparisons with similar networks in other settings. What are the key features of this network and how does it differ from others regarding management, staffing, aims, structure, and the way how it integrates academy with clinical care.

Response: RFPC was better contextualized, and its benefits made clearer for research and healthcare. See lines 89 to 102

Comment: Introduction: I is claimed that they started some ten years ago, and such networks take time to develop. It would be valuable to get some description of the time line, and important milestones.

Response: RFPC’s milestones were included in the introduction, as suggested. See lines 103 to 110. 

Comment: Methods :Page 9-10 incl Fig 2. This description of the population is very valuable but should come earlier, preferably first in the methods section directly as setting or population. It is also possible to place it in the introduction together with a more comprehensive description of the network.

Response: The information on the population was transferred to the first part of the methods section, as suggested. See lines 143 to 146. A more comprehensive description of the network was provided in the Introduction. See lines 89 to 102. 

Comment: Methods/results: No statistical measure of uncertainty has been applied, e.g. adding confidence intervals on proportions and comparisons. Overall I cant see that many comparisons between groups. Perhaps this is wise given the small numbers, but it might have been interesting to see differences in responses based on respondent categories (with an appropriate statistical test)

Response: The aim of our study was to provide a descriptive and exploratory analysis. We did not attempt to compare groups due to the heterogeneity of categories and varying numbers of professionals. While group comparisons may be intriguing, we do not believe they are relevant for the overall analysis. Our focus is on tailoring RFPC for the entire community, rather than just one specific group.

Comment: Results: The initial sentence “questionnaire was accessed by”, does it mean that this number was invited?

Response: The questionnaire was distributed to all clinical research professionals at Fiocruz. Out of the 242 professionals who clicked on the link to access the questionnaire, 122 provided valid responses. This information is now documented in the Methods and Results section. See lines 181, 238 and 239. 

Comment: Results: There are many illustrations – in total 6 figures and 5 tables. I don’t know the maximum allowed number by the journal, but suggest some could be moved to apêndices

Response: Figure 4, Table 4 and Table 5 were moved to Supplementary Material, as suggested. The article has now 5 figures and 3 tables. 

Comment: Discussion: The structure could be improved through adding a first sharp paragraph about the key findings, then more integration between the assessment of own findings in relation to others. Overall, the generalizability and and comparisons with what is known in the scientific literature should be emphasized more to increase the value for the global scientific community.

Response: Discussion was improved as suggested. See lines 341 to 350, 354 to 356, 379 to 381, 416 to 417 and 421 to 423. 

Comment: Discussion: There is a lack of a Strengths and weaknesses section, which is a necessary part of a discussion.

Response: A section on the study’s strengths and limitations section was added, as suggested. See lines 438 to 457.

Comment: There are no subheadings in Abstract (Intro, Aim, Methods, Result, Discussion). I don’t know whether it is required by the journal or not, but think it would add value.

Response: PLOS One guidelines do not require subheadings in the Abstract, but this section was amended to make the text clearer for the reader.

Comment: Texts: Some subheaders have different sizes. There is also some unnecessary repetition in the text (e.g. the response rate that comes twice) and quite a lot of numbers presented in double both in a table and in the free text. The results and discussion parts could therefore be revised and shortened slightly. I also recommend some statements to be modified to emphasize they are perceived by responders.

Response: The subheadings have been arranged in accordance with the journal's guidelines. In order to reduce redundancy of numbers in both the text and tables, we have condensed the results section as recommended.

Comment: Table 5 –Abbreviations should be explained in footer

Response: Table 5 has been moved to the supplementary material and the abbreviations have been included in the footer as suggested

---

## [Editor Report · Decision Letter 1]

12 Jul 2024

Strengthening research networks: insights from a clinical research network in Brazil

PONE-D-23-39952R1

Dear Dr. Fonseca,

We’re pleased to inform you that your manuscript has been judged scientifically suitable for publication and will be formally accepted for publication once it meets all outstanding technical requirements.

Kind regards,

Claudia Garcia Serpa Osorio-de-Castro, Ph.D

Academic Editor

PLOS ONE
---

## [Editor Report · Acceptance letter]

24 Jul 2024

PONE-D-23-39952R1 

PLOS ONE

Dear Dr. Fonseca, 

I'm pleased to inform you that your manuscript has been deemed suitable for publication in PLOS ONE. Congratulations! Your manuscript is now being handed over to our production team.

Kind regards, 

on behalf of

Dr. Claudia Garcia Serpa Osorio-de-Castro 

Academic Editor

PLOS ONE